# Towards Continuing Interprofessional Education: Interaction patterns of health professionals in a resource-limited setting

**Champion N. Nyoni** [ID]*◎, **Cecilna Grobler**◎, **Yvonne Botma**◎

School of Nursing, University of the Free State, Bloemfontein, South Africa

◎ These authors contributed equally to this work.
* nyonic@ufs.ac.za

## Abstract

There are challenges related to collaboration among health professionals in resource-limited settings. Continuing Interprofessional Education initiatives grounded on workplace dynamics, structure and the prevailing attitudes and biases of targeted health professionals may be a vehicle to develop collaboration among health professionals. Workplace dynamics are revealed as health professionals interact. We argue that insights into the interaction patterns of health professionals in the workplace could provide guidance for improving the design and value of CIPE initiative. The study was conducted through rapid ethnography and data were collected from non-participant observations. The data were transcribed and analysed through an inductive iterative process. Appropriate ethical principles were applied throughout the study. Three themes emerged namely "Formed professional identities influencing interprofessional interaction", "Diversity in communication networks and approaches" and "Professional practice and care in resource limited contexts". This study revealed poor interaction patterns among health professionals within the workplace. These poor interaction patterns were catalyzed by the pervasive professional hierarchy, the protracted health professional shortages, limited understanding of professional roles and the lack of a common language of communication among the health professionals. Several recommendations were made regarding the design and development of Continuing Interprofessional Education initiatives for resource-limited settings.

## Introduction

There are challenges associated with collaboration among health professionals in resource-limited settings and current continuing professional development (CPD) programmes seem not to address specific issues related to collaboration. Popular CPD programmes for health professionals, driven by employers or professional bodies, are usually provided for specific professions and aimed at the transfer of knowledge and skills from experts and are seldom delivered in the workplace [1–3]. Zwarenstein et al. [4] demonstrated the potential value to healthcare outcomes of practice based interprofessional education interventions through a systematic review of studies from high-come countries. Practice based CPD activities integrate and

**Funding:** The author(s) received no specific funding for this work.

account for the complex workplace environment which is often missed in pervasive CPD activities in low-resource settings [5].

The workplace environment for health professionals is inherently complex, dynamic, non-linear and often difficult to predict [6]. The ageing population, changes in population health-care needs, epidemiological transitions, resource limitations, and the protracted health work-force shortage contribute to the labyrinth of the workplace environment, which often compromises healthcare [7]. The absence of true collaboration among health professionals, especially in resource-limited workplace settings, worsens the situation [8]. Various scholars and organisations advocate for collaboration among health professionals to enhance person-centered care, improve health professional job-satisfaction and improve systems for health [9].

Collaboration among health professionals in the workplace may not occur spontaneously, especially in cases where their initial training was through uni-professional programmes [10]. Key strategic elements such as careful planning, leadership, and education, need to inform workplace initiatives that foster the development of collaboration among health professionals. Continuing Interprofessional Education (CIPE) reflects deliberate initiatives aimed at improv-ing collaboration among health professionals and enhancing the quality of healthcare within the complex workplace environments through engaging two or more health professionals from different professional backgrounds in learning with, from and about each other [11]. The CIPE initiatives are primarily appropriate for health professionals after their initial qualifica-tion and may be an alternative to traditional CPD programmes in fostering collaboration among health professionals [12, 13].

Veerapen et al. [14] emphasise that the CIPE initiatives should be developed with a clear understanding of workplace dynamics, structure and the prevailing attitudes and biases of tar-geted health professionals underpinned by sound theoretical frameworks. The social identity theory, reflective and experiential learning theory, and learning within communities of prac-tice are proximal theories that can be infused in the design and delivery of the CIPE initiatives [15]. In applying these theories in practice, Owen et al. [15] describes that the CIPE initiatives should have a clear focus on hands-on interprofessional exchange situated in the workplace through interaction and co-participation of all health professionals. The unique aspects of each workplace and associated behaviours of health professionals influence the effectiveness of the CIPE initiatives [2, 15].

Resource-limited settings present unique workplace environments for health professionals. Such workplace environments are generally characterised by limited numbers and types of health professionals, heavy disease burden, and limited operational resources which affect the delivery of healthcare [16]. Collaboration among health professionals, especially in resource-limited settings, would significantly contribute to improved healthcare and health outcomes. Collaboration is valued as essential in such healthcare crises situations such as the COVID-19 pandemic, where healthcare workers could collaboratively support each other and become each other's buffer to burn out. The CIPE initiatives are therefore essential in these settings and such initiatives may be different from those developed in well-resourced settings. As estab-lished, the CIPE initiatives need to be informed by the workplace dynamics, structure and the prevailing attitudes and biases of targeted health professionals which are revealed as health professionals interact within the workplace [14]. The interaction patterns of health profession-als in resource-limited settings are not known. The majority of the literature from the African context reports interprofessional education for pre-registration programmes and there are limited reports that focus on Continuing Interprofessional Education. We argue that insights into the interaction patterns of health professionals in the workplace could provide guidance for improving the design and value of CIPE initiatives as a vehicle towards collaboration among health professionals in resource-limited settings. Resource limited settings lack funds

that affect individuals, the society and/or institutions to access healthcare. In resource limited settings, there is limited access to medication, less-developed infrastructure, few-numbers of trained health professionals and poorly maintained infrastructure. Therefore, this article aims to describe the interprofessional interaction patterns of health professionals in the workplace in a resource-limited setting.

## Context of this study

South Africa, with a population of over 51 million, is plural society evidenced by over ten official languages and multiple cultures. Cultural nuances laced with patriarchy and bigotry influence the interactions of individuals in the society and in organised structures such as healthcare environments. Healthcare environments in South Africa include the private, the public and the traditional/cultural health care system. Eighty-four percent (84%) of the population do not have health insurance and access the public healthcare system which receives only half of the national health resources [17]. The public health system is underpinned by the primary healthcare philosophy presented through a tiered pathway for patients influenced by the complexity of their healthcare needs. Central hospitals are at the pinnacle of this system, where advanced specialised care is expected [17]. However, these hospitals are often poorly funded, with poorly maintained infrastructure, few numbers of professionals, limited medications and overcrowding [16].

This study was conducted at a conveniently selected central hospital. This hospital received referral patients from a wide geographic area including patients referred directly from a neighbouring country. In as much as there is a diversity of health professionals within this setting, their numbers are not enough [17]. Patients predominately communicate in one of the indigenous languages which is predominant in the region, while health professionals use English as a medium of communication. This central hospital reflects challenges similarly faced by most central hospitals in the country which include, lack of funding, poorly maintained infrastructure, shortage of consumables, shortage of health professionals and overburdened health system. Additionally, this setting has fragmented health services, limited health technologies, dysfunctional health communication technology, and a heavy disease burden. These challenges have not received systemic attention and have influenced the quality of healthcare and health indicators [16].

## Methods

This study was approved by the Health Sciences Research Ethics committee of the University of the Free State (HSD2018/0055/3007) and the Department of Health (Free State Province) in South Africa. All participants provided written consent to be part of the study.

## Design

This qualitative research study was conducted through rapid ethnography following a non-participant unstructured observation technique [18]. Due to the challenges associated with describing human interaction, rapid ethnography allowed for the direct examination and observation of the interaction of health professionals in real-time. In addition allowing for the observation of in-depth information on some topics that participants may be unwilling to talk about, unaware of or even unable to recall especially in this setting where cultural nuances may influence the ability of the participants to express their experiences [18, 19].

## Study setting and unit of analysis

Four wards, within this hospital, were purposively selected to be part of this study, as they reflected ideal opportunities for the interaction of health professionals. These four wards served a similar population group with diverse health conditions and each ward had a bed capacity of thirty and were managed by different health professionals inclusive of medical doctors, specialist doctors, nurses, physiotherapists, nutritionist, and speech therapist. In this context, caregivers who are parents or next of kin are part of the care of these patients, and are admitted into these wards with the patients. An example of such an ideal opportunity was an admitted patient with a complex health condition, with their caregivers present, and managed by multiple health professionals of different professional backgrounds.

## Data collection

The data were generated through non-participant unstructured observations of the interaction of health professionals at four wards in a central hospital in South Africa [18]. Two of the authors collected the data (CNN, CG). One of the data collectors (CG) had extensive experience of living in and understanding cultural nuances in the setting, while the other was a professional nurse (CNN), and additionally, both had extensive experience in qualitative research. The data were collected over a three-week period guided by a data collection schedule and data saturation. The data collection schedule allowed for opportunistic and planned opportunities for observation at different times of the day throughout the four wards [18]. During data collection, the researchers only observed cases where there was interaction among two or more health professionals from different professional backgrounds during patient care. In addition, the data collectors also observed the influence of the physical environment on the interaction of health professionals. Data were captured as field notes and reflection notes. Using the transforming learning theory by Mezirow [20] focusing on the critical examination of the observations and experiences in the wards, the researchers discussed their observations immediately after the data collection, interpreted the observations and used the outcomes of the discussions to guide subsequent data collection and saturation. Saturation was reached when the researchers were not generating new information from the reflections [21].

## Data analysis

The analysis of the data was executed in three sequential steps [22]. The initial steps focused on the process of organising the collected data and preparing such data for in-depth analysis. This step involved the transcription of all data collected from observations and reflections using an author generated transcription guideline. The transcription guideline influenced the formatting of each transcript. Each transcript was then allocated a unique code based on the day of the data collection, the time when the data were collected and the ward in which the data were collected. The days were numbered numerically from the first day of data collection with the first day labelled as Day 1, while the time was labelled based on whether the data were collected in the morning (AM) or afternoon (PM) and lastly, the last four letters of the alphabet were used to label the four wards were the data was collected.

The transcripts were then stored in a folder on Atlas.ti [TM] for the second step of the data analysis process. The second step of the data analysis process also referred to as the first cycle of data analysis, was conducted through an inductive approach applying several coding methods including, open, structural, axial and in-vivo coding methods [22]. The generated codes from all the manuscripts were then analysed in the third step of the data analysis process or the second cycle of data analysis through pattern coding [22]. Pattern coding is an approach of grouping summaries of data, or codes into themes, based on inherent code patterns. The

outcome of this analysis were themes that are presented with supporting statements from the collected data.

## Ethics

This study is part of a larger study aimed at developing a Continuing Interprofessional Education Short learning programme for health professionals in paediatric settings in Sub-Saharan Africa. The study applied the Department of Health (South Africa) ethics framework, which is underpinned by fundamental research ethics principles [23]. All participants in this study provided written consent to be part of this study, after an in-depth description of the study and data collection approaches.

**Rigor of study.** The quality criteria by Richardson [24] guided this rapid ethnographic study. This study made a substantive contribution to the understanding of interaction patterns by health professions in low-resource setting. Regarding aesthetic merits, appropriate analytical approaches were used in this study and also in the presentation of the outcomes of this research. Reflexivity, including appropriate ethics behaviour and awareness of the authors' inherent biases were taken to account in the collection and interpretation of the data. The verbatim quotes from the field notes have been used to support the credibility of interaction patters. Specific recommendations for action have been presented.

## Findings

The interaction patterns of health professionals at a central hospital in South Africa are presented as three themes namely: Formed professional identity, Communication networks and approaches; and Professional practice and care. The themes of the study are discussed and supported with evidence from the data.

### Formed professional identity influencing interprofessional interaction

The formed professional identities, anchored in professional hierarchy, were reflected in the overt interactions and behaviours of the health professionals throughout the wards. Leadership and final decision-making regarding all aspects of patient care seemed to be an expected professional role of the Doctors. Such final decisions appeared to extend beyond the Doctors' scope or professional boundaries such as prescribing specific physical therapy interventions in the presence of a qualified Physiotherapist. These interventions seemed to be prescribed in a non-negotiable atmosphere with an expectation of such a prescription to be duly executed. Some of the other health professionals' roles seemed to be limited to executing the "Doctor's orders" against a negative consequence when not done as prescribed. In one of the observations, it was clear that;

> "...the Doctor seems very annoyed because the other health professional did not follow his prescription" (Day 4, Ward X, AM)

The initiation, leadership, and coordination of a ward round were at the discretion of the Doctors, who were expected to decide when and how such ward rounds should be conducted. The researchers had scheduled to observe the interactions of health professionals in one of the wards included in this study, with a reputation of a weekly multi-professional ward round. However, despite the presence of other health professionals on the scheduled time, the round could not proceed as scheduled as the Doctor that normally initiated, led and coordinated the round was said to be on leave.

*"After 40 minutes of waiting for the consultant Doctor, all the health professionals had to disperse. It was reported that the consultant was on leave. . .it seems the ward round cannot continue without the consultant Doctor" (Day 2, Ward X, AM).*

In addition to the expected leadership role of the Doctor, other health professionals such as Nurses seemed to deliberately disengage with the decision-making processes related to patients under their care. The role of the Nurses in the rounds seemed to be limited to accompanying the Doctors, answering a few questions related to the execution of Doctors' orders and translating Doctors' decisions on patient care to patients and caregivers, especially in cases where the language was a barrier for effective communication. During one observation:

*"The Doctor requests for specific laboratory results. . .the nurse brought the result. . .when asked specific questions on the result, she responds in brief sentences and immediately leaves. . ." [Day 3, Ward Y, PM]*

## Diversity in communication networks and approaches

Multiple languages were used to verbally communicate messages among health professionals while English was the predominant language of communication in written documents. The choice of language for verbal communication seemed to be influenced by health professionals. In most cases, health professionals would communicate directly with caregivers in the caregivers' own language, which seemed to invoke some increased participation and involvement in the discussions. In cases where a health professional could not speak the language of the caregiver, Nurses were used as conduits for communication, as Nurses seemed to understand multiple local languages. However, using the Nurses for communication seemed to have its limitations as messages appeared not to be passed immediately and some information was lost in translation.

*". . .as the caregiver sobbed bitterly, the Doctor asks the nurse to explain why she will not be discharged. The nurse explains that she will do it after the ward round. . ." [Day 6 Ward W, AM]*

The communication networks involved several role players namely the Doctors, Nurses, other health professionals and caregivers to the admitted patients. It is important to note that the ward round seemed to continuously comprise Doctors and Nurses as significant role players, with the caregivers in the periphery and other health professionals occasionally joining in for a specific question or discussion with the Doctors. Other health professionals such as the Physiotherapist would engage in a direct conversation with the Doctor, excluding the Nurse.

*"The Physiotherapist enters and talks to the caregivers and makes notes. . .she [physiotherapist] leaves the room and quickly returns to interrupt the Doctor and Nurse round for a quick word with the Doctor, then she leaves. . ." (Day 3, Ward Z, PM)*

Documents were central to the communication network among health professionals. Doctors seemed to continually review and write out their specific management approaches within patient notes. The writing on patient notes was consistent throughout all patients included in the ward round. Documentation appeared not to be accompanied by discussions and other health professionals would be referred to the written notes and added on to the notes without verbally communicating their actions and notes to the health professionals in the ward.

*". . .the Doctor continues to write. . .and as they move to the next patient the Physiotherapists, accesses the notes, reads the notes and seems to be adding on to the notes. She leaves without communicating what she has written on the notes. . ." [Day 2, Ward Y, AM]*

In addition to the language and documentation, the demeanour of the health professionals appeared to be a significant part of communication approaches. The Doctors seemed to have poise when communicating with other health professionals and caregivers, reflecting a superior status. The other health professionals cemented the Doctors' striking status within the communication network by displaying subservient demeanour. The Nurses preferring appeared to be backbenchers when engaging with other health professionals exhibiting gestures that reflected non-engagement inpatient care discussions. The same Nurses, however, would be overly animated when communicating with caregivers.

*"The nurse seems very bored with the discussion in this round, she is constantly looking through the window at events outside the hospital while Doctors are discussing patient care. She immediately reaches for one of the caregivers' chairs and takes a seat. . .and reaches towards her cell phone" [Day 1, Ward X, AM]*

## Professional practice and care within resource limited contexts

The perceived professional roles of other health professionals influenced professional practice and the quality of healthcare. Health professionals seemed to be providing independent healthcare on the same patient and caregiver. Different health professionals would engage the same caregiver at different times of the day for similar health information as part of patient care. Conversely, the caregivers seemed fatigued by the iterative nature of the inept approach to healthcare.

*"The caregiver explains that she had answered similar questions from the other health professional, and does not understand why she should answer these questions again" [Day 5, Ward W, AM]*

During data collection, it became apparent that Nurses spent a large amount of time and focus within the corridors of the wards and not with the patients and caregivers. It seemed as if the Nurses had, to an extent, relinquished some of their professional roles to the caregivers and at the same time it was not clear the exact model of nursing care being applied within the included wards. Eventually, the quality of nursing care may be compromised when the Nurses are distant from the patients under their care.

*"The cubicles only have patients and their caregivers. All the Nurses are in the corridors engaged in different activities. . ." [Day 5, Ward Y, AM]*

The physical environment of the wards contributed to the quality of professional practice and care. The patients and their caregivers were housed inwards that were structured in the form of cubicles. Each cubicle had four beds that accommodated the patients and then corresponding chairs for the caregivers. Ward rounds, including the discussion of patient's sensitive and private issues, were conducted in the vicinity of other present caregivers in the cubicle. This practice compromised the patients' ethical rights to privacy and dignity.

*"As the round commences, the discussion on the outcomes of the patient's surgery is made in front of all patients in the cubicle. There are no partitions to separate these patient beds." [Day 3, Ward W, AM]*

The caregivers and the patients seemed to have limited influence on the decisions of their own care. During decision-making, the caregivers were only limited to responding to specific questions related to their health. In some cases, it seemed as if the caregivers' own needs were not met or were deliberately ignored especially during the ceremonious ward round. Health professionals would further pledge to attend to the patient and caregivers' needs at the end of the ward round.

*"After being told that she (caregiver) would not be going home, she cried uncontrollably. . .the health professionals continued with the round. The nurse told her, she will come and talk to her after the round" [Day 6, Ward W, AM]*

## Discussion

The purpose of this study was to describe the interprofessional interaction patterns of health professionals in a resource-limited setting. Understanding the interprofessional interaction patterns of health professionals would support the development of CIPE initiatives that are influenced by the workplace dynamics, structure and prevailing attitudes and biases of targeted health professionals. This study provides a glimpse of how health professionals in a typical low-resource setting in South Africa seemed to interact. The study outcomes seem not to be unique to low-resource settings, as earlier studies in higher income countries have demonstrated similar outcomes [25, 26]. Contextual, processual, relational and organisational factors seem to heavily influence the interprofessional interaction patterns in this setting [27].

Professional identity is a concept associated with how professionals perceive themselves within an occupational context and in what way such a perception is revealed to others [28]. The infrastructure, education, socialisation, and society, contribute to professional identity formation and such identities may not be aligned to the professional roles and scope as described by regulators. The professional identity of the health professionals in this study seemed to mirror professional hierarchy, with Doctors at the apex of this hierarchy and the other health professional fitting at its base. It appeared as if Doctors were expected to naturally lead patient care and give directives to all other health professionals who seemed to have a reporting role. Professional hierarchy in the workplace for health professionals has been reported in various settings [29] with debilitating patient outcomes, poor job satisfaction and tardy decision-making processes [30]. The omnipresent patriarchal culture, wide salary variation, and difference in the level of education of the health professionals in this setting, which are described by Reeves et al. as contextual factors [27], seem to rivet professional hierarchy among health professionals.

CIPE initiatives underpinned by transformative learning would be valuable for this setting. The patterns of interprofessional interaction seem to be influenced by deeply seated worldviews originating from the "habits of the mind" influenced by family, culture and the community [31]. Mezirow describes transformative learning as aimed at influencing the world views of adult learners through critical reflection on own beliefs, assumptions and perspectives [20, 32]. Initially, health professionals would benefit from facilitated learning on self-reflection and self-knowledge. Self-reflection is fundamental to transformative learning, as it empowers an individuals to question the "habits of the mind". Understanding the professional roles of other health professionals nested within the social identity theory, would support health professionals in recognising their identities and roles within an occupational group that has shared knowledge and values [15]. Case studies, simulations, and practical interprofessional exchanges underpinned by experiential learning principles would allow health professionals as a team to engage and reflect on their various professional roles within patient care. Sweeny

et al. [33] further explain that the flattening of professional hierarchies should be aligned with a leadership approach that encourages shared decision making, value clarification, professional autonomy and professional accountability. Such a leadership approach when executed in the workplace may steer the focus of decision making to rest on all health professionals instead of the Doctor [34].

Nurses and Doctors seemed to be the key role players with the communication networks recognised in this study, with caregivers and the other health professionals peripherally engaged. The chronic shortage of health professionals, especially the allied health professionals in resource-limited settings, justifies the predominant role players in patient care in this study. This chronic shortage of allied health professionals has been attributed to various factors including limited training opportunities, poor regulation and limited utilisation in resource-limited settings [35]. The shortage of health workers may not necessarily justify the observed interprofessional interaction patterns, however CIPE initiatives in such settings must be sensitive to the shortage of health professionals through integrating trans-professional models of care. Trans-professional models of care are aimed at increasing the range of services offered by health professionals through flexibility in professional boundaries, workload distribution and support during problem-solving [36, 37]. Trans-professional models of care may need to be carefully applied and be within the remit of professional regulation. Trans-professional models of care must be driven by patient needs and aimed at improving patient outcomes through engaging patients and their caregivers in their own care [37]. Trans-professional care could enhance efficiency in the utilisation of health professionals, creating high performing teams that would stretch the limited resources. Such a model of care will enhance health professionals to support each other within a resource-limited context.

In multi-professional models of care, health professionals from different professional backgrounds work in parallel to perform professional specific care such as conducting a profession-specific assessment, planning, and care [37]. Caregivers in this study were exposed to this model of care, which seemed iterative and irritating, as similar questions for example during history taking were asked by different health professionals. The CIPE initiatives should emphasise the development of a common language of communication among the health professionals, which would lead to interdependence in health interventions. The International Classification of Function, Disability and Health (ICF) framework proposed by the World Health Organisation (WHO) may be used in structuring approaches towards a common language of communication among health professions. The ICF has been reported to catalyse interprofessional exchange, further improving collaboration among health professionals, especially in resource-limited settings, where multi-professional models of care may be expensive and resource-intensive [38].

True collaboration among health professionals as an outcome of the CIPE initiatives needs to involve the patients and caregivers in decisions regarding their own care. Patients and caregivers in this setting seemed to have limited roles in their own care and this may have been a result of the pervasive culture of traditional authority, with patients and caregivers subservient to health professionals. Ward et al. [39] explained that involving patients in decision making regarding their own care has direct benefits which include improved compliance to management approaches. Dismantling subservience of patients and caregivers may be a challenge as such cultures are part of social structures which expect authority to be respected. The CIPE initiatives should aim at the collaboration of health professionals towards person-centered respectful care that acknowledges the context of the patients and caregivers.

Typical of resource-limited settings is the physical environment of the workplace of health professionals which presents a challenge for engaging in true collaboration and the provision of quality care. This study revealed how the environment influenced fundamental

patient care issues such as confidentiality due to the absence of individualised compartments for patient care. Some workplace environments in these resource-limited settings may not accommodate a large interprofessional team to provide direct care to patients while in some case there may not be space for meetings and dialogue which are necessary for collaboration. The use of shared documents, which allow health professionals to communicate in a common language may be an alternative to resource-limitations that arise to the challenges of space.

## Conclusion

The collaboration of health professionals within the workplace has been linked to improved patient care, patient outcomes and improved healthcare worker satisfaction [9]. Health professionals in resource-limited settings are not trained to work collaboratively as the majority of undergraduate training is provided in uni-professional programmes. Current CPD activities for health professionals usually do not address the collaboration of health professionals and are conducted away from the complex workplace environments.

CIPE initiatives situated within the complex workplace environment may be engaged in developing collaboration among health professionals. These initiatives need to be designed based on the contextual underpinnings of their intended beneficiaries. From this study, the following are recommended to designers of CIPE initiatives in resource-limited settings;

- The CIPE initiatives should be underpinned by transformational learning philosophies linked with practical interprofessional exchanges among health professionals in the workplace;

- Engage distributive leadership approaches that reflect professional accountability, shared decision making, value clarification, and professional autonomy;

- The CIPE initiatives should be sensitive to human resource shortages and engage in trans-professional models of care driven by patient care needs;

- There should be a common language of communication among health professionals through the use of the ICF framework;

- There should be involvement and further research around culturally sensitive patient involvement in health care; and

- A shared documentation process that is driven by a common language.

The authors acknowledge limitations inherent within this study and the design applied, where further triangulation of data could have strengthened the authors' perceptions of the interaction patterns of the health professionals. However, the data generated from this study is crucial in providing insights into the interprofessional interaction patterns of health professionals in resource-limited settings which seems to confirm research conducted in high-income settings. It is recommended that further research in this field should engage various strategies to further promote true collaboration among health professionals in resource-limited settings.

## Supporting information

**S1 File.**
(DOCX)

**S2 File.**
(DOCX)

**S3 File.**
(DOCX)

**S4 File.**
(DOCX)

**S5 File.**
(PDF)

## Acknowledgments

The authors would like to acknowledge Prof. Ruth Albertyn for critical insights in the structuring of this article.

## Author Contributions

**Conceptualization:** Champion N. Nyoni, Cecilna Grobler, Yvonne Botma.

**Data curation:** Champion N. Nyoni, Cecilna Grobler, Yvonne Botma.

**Formal analysis:** Champion N. Nyoni, Cecilna Grobler, Yvonne Botma.

**Methodology:** Champion N. Nyoni, Yvonne Botma.

**Project administration:** Champion N. Nyoni.

**Supervision:** Yvonne Botma.

**Validation:** Champion N. Nyoni, Cecilna Grobler.

**Writing – original draft:** Champion N. Nyoni.

**Writing – review & editing:** Champion N. Nyoni, Cecilna Grobler, Yvonne Botma.

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
