## [Decision Letter · Decision Letter 0]

13 May 2021

PONE-D-21-07227

Towards Continuing Interprofessional Education: Interaction patterns of health professionals in a resource-limited setting

PLOS ONE

Dear Dr. Nyoni,

Thank you for submitting your manuscript to PLOS ONE. After careful consideration, we feel that it has merit but does not fully meet PLOS ONE’s publication criteria as it currently stands. Therefore, we invite you to submit a revised version of the manuscript that addresses the points raised during the review process.

The Academic Editor has served as the second reviewer and and agree with the assessment that minor revisions  are needed. The authors are asked to specifically address the points made by the reviewer.

We look forward to receiving your revised manuscript.

Kind regards,

Joseph Telfair, DrPH, MSW, MPH

Academic Editor

PLOS ONE

Journal Requirements:

Reviewers' comments:

Reviewer's Responses to Questions

**Comments to the Author**

1. Is the manuscript technically sound, and do the data support the conclusions?

Reviewer #1: Yes

2. Has the statistical analysis been performed appropriately and rigorously? 

Reviewer #1: N/A

3. Have the authors made all data underlying the findings in their manuscript fully available?

Reviewer #1: No

4. Is the manuscript presented in an intelligible fashion and written in standard English?

Reviewer #1: Yes

5. Review Comments to the Author

Reviewer #1: This is a well-written paper with interesting topic. Several comments for the further improvement of the manuscript:

1. Please tell the readers more about the resource-limited situation as mentioned in the title. We as the reader might learn more from your paper if we get the picture of the context that you are presenting.

2. You have performed systematic three-step analysis. Please tell the reader more about the process of how you got into the three themes as mentioned in the finding. Visual presentation e.g. table or figure is recommended.

Good luck.

6. PLOS authors have the option to publish the peer review history of their article (what does this mean?). If published, this will include your full peer review and any attached files.

Reviewer #1: No

---

## [Author Response · Author response to Decision Letter 0]

31 May 2021

The authors have described the context of this study in the heading “context of the study” which generally describes the health delivery system in South Africa and also in the hospital included in this study. We have added an additional sentence to the manuscript to enhance clarity on the picture of the resource limitation of the study.

The process of data analysis was described extensively under the section of data analysis including how each step of data analysis was executed including the coding methods for data analysis as supported by Saldana (2016). We have also added a sentence that explains pattern coding which underpinned the theme development process of this study. We are also conscious of the word count, limit and structure of the paper to add more.

Thank you, your comments were insightful.

---

## [Decision Letter · Decision Letter 1]

7 Jun 2021

Towards Continuing Interprofessional Education: Interaction patterns of health professionals in a resource-limited setting

PONE-D-21-07227R1

Dear Dr. Nyoni,

We’re pleased to inform you that your manuscript has been judged scientifically suitable for publication and will be formally accepted for publication once it meets all outstanding technical requirements.

Kind regards,

Joseph Telfair, DrPH, MSW, MPH

Academic Editor

PLOS ONE

Additional Editor Comments (optional):

The Academic Editor served as the second Reviewer and agree the manuscript has addressed all concerns and is acceptable for publication.

Reviewers' comments:

Reviewer's Responses to Questions

**Comments to the Author**

1. If the authors have adequately addressed your comments raised in a previous round of review and you feel that this manuscript is now acceptable for publication, you may indicate that here to bypass the “Comments to the Author” section, enter your conflict of interest statement in the “Confidential to Editor” section, and submit your "Accept" recommendation.

Reviewer #1: All comments have been addressed

2. Is the manuscript technically sound, and do the data support the conclusions?

Reviewer #1: Yes

3. Has the statistical analysis been performed appropriately and rigorously? 

Reviewer #1: N/A

4. Have the authors made all data underlying the findings in their manuscript fully available?

Reviewer #1: Yes

5. Is the manuscript presented in an intelligible fashion and written in standard English?

Reviewer #1: Yes

6. Review Comments to the Author

Reviewer #1: (No Response)

7. PLOS authors have the option to publish the peer review history of their article (what does this mean?). If published, this will include your full peer review and any attached files.

Reviewer #1: No

---

## [Editor Report · Acceptance letter]

1 Jul 2021

PONE-D-21-07227R1 

Towards Continuing Interprofessional Education: Interaction patterns of health professionals in a resource-limited setting. 

Dear Dr. Nyoni:

I'm pleased to inform you that your manuscript has been deemed suitable for publication in PLOS ONE. Congratulations! Your manuscript is now with our production department. 

Kind regards, 

on behalf of

Dr. Joseph Telfair 

Academic Editor

PLOS ONE